# Genome Editing to Generate Sake Yeast Strains with Eight Mutations That Confer Excellent Brewing Characteristics

**DOI:** 10.3390/cells10061299

**Published:** 2021-05-24

**Authors:** Tomoya Chadani, Shinsuke Ohnuki, Atsuko Isogai, Tetsuya Goshima, Mao Kashima, Farzan Ghanegolmohammadi, Tomoyuki Nishi, Dai Hirata, Daisuke Watanabe, Katsuhiko Kitamoto, Takeshi Akao, Yoshikazu Ohya

**Affiliations:** 1Department of Integrated Biosciences, Graduate School of Frontier Sciences, The University of Tokyo, Chiba 277-8562, Japan; 8715654709@edu.k.u-tokyo.ac.jp (T.C.); ohnuki@edu.k.u-tokyo.ac.jp (S.O.); mao.kashima@mcls-ltd.com (M.K.); 2344785075@edu.k.u-tokyo.ac.jp (F.G.); 2National Research Institute of Brewing, Higashi-Hiroshima, Hiroshima 739-0046, Japan; isogai@nrib.go.jp (A.I.); t.goshima@nrib.go.jp (T.G.); akao_t@nrib.go.jp (T.A.); 3Sake Research Center, Asahi Sake Brewing Co. Ltd., Nagaoka, Niigata 949-5494, Japan; nishitomoyuki@asahi-shuzo.co.jp (T.N.); dhirata@agr.niigata-u.ac.jp (D.H.); 4Sakeology Center, Ikarashi Campus, Niigata University, Niigata 950-2181, Japan; 5Department of Molecular Biotechnology, Graduate School of Advanced Sciences of Matter, Hiroshima University, Higashi-Hiroshima, Hiroshima 739-8530, Japan; 6Graduate School of Agriculture, Kyoto University, Uji, Kyoto 611-0011, Japan; watanabe.daisuke.2w@kyoto-u.ac.jp; 7Department of Pharmaceutical and Medical Business Sciences, Nihon Pharmaceutical University, Bunkyo-ku, Tokyo 113-0034, Japan; k-kitamoto@nichiyaku.ac.jp; 8Collaborative Research Institute for Innovative Microbiology, The University of Tokyo, Tokyo 113-8657, Japan

**Keywords:** sake yeast, genome editing, *MDE1*, *CAR1*, *FAS2*, CRISPR/Cas9

## Abstract

Sake yeast is mostly diploid, so the introduction of recessive mutations to improve brewing characteristics requires considerable effort. To construct sake yeast with multiple excellent brewing characteristics, we used an evidence-based approach that exploits genome editing technology. Our breeding targeted the *AWA1*, *CAR1*, *MDE1*, and *FAS2* genes. We introduced eight mutations into standard sake yeast to construct a non-foam-forming strain that makes sake without producing carcinogens or an unpleasant odor, while producing a sweet *ginjo* aroma. Small-scale fermentation tests showed that the desired sake could be brewed with our genome-edited strains. The existence of a few unexpected genetic perturbations introduced during breeding proved that genome editing technology is extremely effective for the serial breeding of sake yeast.

## 1. Introduction

Sake is an alcoholic beverage that originated in Japan and has been made for centuries. It is made by parallel double fermentation of sake rice with a high starch content, mainly by the action of two types of microorganisms [1]. The microorganisms inoculated are *Aspergillus oryzae* and the yeast *Saccharomyces cerevisiae*. Genome sequence analysis has revealed that sake yeast is a different clade (i.e., subspecies) than wine or ale brewer’s yeasts [2]. Sake yeast also originated in Japan and is specifically tolerant to high alcohol concentration and low pH, with a high fermentation ability for making sake with excellent flavor characteristics. These characteristics indicate that sake yeast has been domesticated for a unique purpose in a geographically isolated situation.

According to recent statistics from Japan’s National Tax Agency, more than 1400 sake breweries exist in Japan and more than 10,000 sake brands are sold globally. The variety in sake brewing arises from the existence of many kinds of sake yeast isolates that have been bred by the National Research Institute of Brewing, prefectural brewing institutes, and private research laboratories in sake breweries [3]. Some of the isolates developed by the National Research Institute of Brewing are known as kyokai yeast and are distributed by the Brewing Society of Japan. To begin development, excellent yeasts suitable for sake brewing were selected from the yeasts used in sake breweries. Then, “parallel” or “serial” breeding techniques were used to meet social demands and fulfill consumer taste expectations. An example of parallel breeding output is non-foam-forming yeast. The excellent sake yeasts that were previously used in sake breweries caused foam to accumulate in the upper layer of the fermentation tank. However, this foam production creates a need for large tanks and increases the labor requirements among sake brewery workers. In the breeding selection method developed by Ouchi and colleagues [4], non-foam-forming yeasts were bred in parallel from several excellent yeast isolates: non-foam-forming kyokai strain numbers 601 (hereafter designated K601), K701, K901, K1001, and K1401 were bred from strains K6, K7, K9, K10, and K14, respectively [5]. By contrast, serial breeding involves the creation of a single strain with several excellent traits through a successive breeding operation. The non-foam-forming strain K1601 originated from strains K7, K9, and K10, and produces excellent sake that is highly aromatic and slightly acidic. Strain K1601 was used to breed strain K1801, which produces considerable *ginjo* aroma [6]. Strain K1801 was then used to breed strain K1901, which produces less ethyl carbamate, a group 2A carcinogen [7]. Because sake yeast is diploid, breeding efficiency is limited despite the use of appropriate breeding selection methods. Therefore, serial breeding requires long durations of time and considerable effort.

Comparison of the genome sequences of kyokai yeast has revealed many mutations that accumulated during the breeding process [2]. The whole-genome sequence of strain K7 was compared with the sequence of the canonical non-foam-forming strain K701. This comparison identified heterozygous and homozygous nonsynonymous variants in 73 and 5 genes, respectively, although the two strains have a parent–child relationship [8]. Strain K701 was isolated in the early 1970s, and both isolates have accumulated mutations in parallel over nearly 50 years. However, unexpected mutations were presumably introduced during the breeding of the canonical non-foam-forming strain. To maintain consistent brewing properties, other than the properties of interest, unexpected mutations should be minimized.

The relationship between yeast genotype and phenotype has been studied by multiple researchers, revealing several genes important for brewing characteristics. Shimoi, et al. found that the non-foam-forming strain K701 has a chromosomal translocation in the *AWA1* region that encodes the mannoprotein present in the cell wall [9]. The hydrophobicity of the cell surface is altered by the loss of *AWA1* activity, such that carbon dioxide is no longer bound to the cell surface, leading to the non-foam-forming phenotype [9]. Kitamoto, et al. revealed that the loss of the arginase gene (*CAR1*) results in reduced urea production, thereby eliminating the potential carcinogen ethyl carbamate derived from urea that forms during the storage of sake [7]. Ichikawa, et al. found that sake yeast strains producing the strong *ginjo* aroma of ethyl caproic acid have the Gly1250Ser mutation in the *FAS2* gene [10], which encodes the alpha subunit of fatty acid synthase. This dominant *FAS2* mutation was thought to decrease fatty acid synthesis, resulting in decreased carbon chain elongation and increased amounts of caproic acid and caproyl-CoA, precursors of ethyl caproate [10]. Furthermore, sake yeast defective in the *MDE1* gene reportedly exhibits reduced production of dimethyl trisulfide (DMTS), a strong, unpleasant odor known as *hineka* [11]. Evidence-based genetic engineering based on genotype–phenotype relationships enables the construction of sake yeast strains with superior properties. However, homozygous alleles must be prepared when introducing recessive mutations and markers must be prepared for selection of the recombinant yeast.

In this study, we attempted to construct sake yeast with excellent brewing characteristics by using genome editing technology. We introduced eight mutations into K7 to construct non-foam-forming sake yeast with three additional excellent brewing characteristics. To construct a non-foam-forming strain that makes sake without producing carcinogens or an unpleasant odor, instead producing a sweet *ginjo* aroma, we introduced the *car1*∆/*car1*∆, *mde1*∆/*mde1*∆, and *FAS2* (G1250S)/*FAS2* (G1250S) mutations into *awa1*∆/*awa1*∆, in that order. A small-scale fermentation test showed that the desired sake was achieved. Based on whole-genome sequencing and morphological analyses of the genome-edited strains, we propose that genome editing technology can be applied for the effective construction of sake yeast strains by serial breeding.

## 2. Materials and Methods

### 2.1. Strains and Media

The sake yeast strains used in this study were the foaming isolate K7 and genome-edited, non-foam-forming isolate K7GE01 [8]. The genome-edited strains constructed in this study were K7GE21, K7GE31, and K7GE41. Yeast strains were cultivated at 30 °C in yeast extract peptone dextrose medium containing 1% (*w/v*) Bacto yeast extract (BD Biosciences, Palo Alto, CA, USA), 2% (*w/v*) Bacto peptone (BD Biosciences), and 2% glucose to transform yeast trains, extract yeast DNA, and prepare precultures for fermentation tests. Geneticin (Takara, Kyoto, Japan) was added at 350 μg/mL to yeast extract peptone dextrose agar plates containing 2% agar (Shouei, Tokyo, Japan) after autoclaving.

### 2.2. Application of Genome Editing Techniques to Sake Yeast Strains

We applied genome editing techniques to the *AWA1*, *CAR1*, *MDE1*, and *FAS2* genes located on chromosomes XV, XVI, X, and XVI, respectively. pCAS-Pro-AWA1 plasmid co-expressing nuclease protein Cas9 and Ribozyme-sgRNA that guides the Cas9 to the target *AWA1* sequence were described in Ohnuki, et al. [8]. We constructed the pCAS-Pro-CAR1 plasmid from pCAS Prolin-URA3 [12] with the *CAR1* target sequence (Appendix A) using a restriction-free cloning method [13]. The repair DNA used had a length of 150 bp (*CAR1* donor sense and *CAR1* donor antisense for K7GE21) and was generated by annealing equimolar amounts of single-stranded oligonucleotides as follows: the mixture was first denatured at 100 °C for 5 min and then allowed to cool to 25 °C with a ramp of 0.1 °C/s [14]. We constructed pCAS-Pro-MDE1 and pCAS-Pro-FAS2 (G1250S) plasmids in a similar manner.

To measure the genome editing efficiency of each sake yeast strain, K7 was co-transformed with the pCAS-Pro-AWA1, pCAS-Pro-CAR1, pCAS-Pro-MDE1, and pCAS-Pro-FAS2 (G1250S) plasmids, as well as the corresponding repair DNAs, using an improved lithium acetate transformation method. To construct strain K7GE21, K7GE01 was co-transformed with the pCAS-Pro-CAR1 plasmid and the corresponding repair DNA. After selection of G418^r^ colonies, we amplified the genomes of the yeast transformants by colony polymerase chain reaction (PCR) (primers described in Appendix A) to confirm the presence of desired fragments. We amplified PCR fragments of 755 and 1757 bp with deleted and intact copies of the *CAR1* gene, respectively. When both deleted and intact copies were detected, we re-examined the heterozygosity after reisolating single colonies. After we confirmed the presence of the *car1*∆/*car1*∆ alleles, we allowed the pCAS-Pro-CAR1 plasmid to be spontaneously eliminated, producing the genome-edited strain, K7GE21. We constructed strains K7GE31 and K7GE41 in a similar manner by using pCAS-Pro-MDE1 and pCAS-Pro-FAS2 (G1250S) plasmids.

### 2.3. Whole-Genome Sequencing

We extracted DNA from strains K7, K7GE01, K7GE21, K7GE31, and K7GE41 to determine their whole-genome sequences. First, we isolated high-molecular-weight DNA (10 µg, ~24 kb fragments) with a Genomic-tip 100/G kit (QIAGEN, Germantown, MD, USA), in accordance with the manufacturer’s instructions. We estimated the purity of the DNA samples using a spectrophotometer (Nano Drop; Thermo Fisher Scientific, Waltham, MA, USA). The whole-genome sequencing was outsourced to GeneBay (Yokohama, Japan). The DNA samples were sent to Novogene (Singapore) to prepare a PCR-free, paired-end, sequencing library and for whole-genome sequence analysis (2 × 150 bp) using an Illumina NovaSeq 6000 sequencing platform (Illumina, San Diego, CA, USA) at ~100-fold nominal coverage. The adapter contamination was removed and the low-quality bases trimmed. We obtained the K7 reference genome (NRIB_SYGD, txid721032) from the Sake Yeast Genome Database (https://nribf1.nrib.go.jp/SYGD/, ver. 1.0) and prepared it for use in sequencing data analysis. The software packages that we used for sequencing data analysis were: Sequence Alignment/Map Tools (ver. 1.11) [15] to convert the “sam” format to “bam” format and to modify information concerning the paired reads; Burrows–Wheeler Aligner (ver. 0.7.17) [16] for mapping reads to the K7 reference genome; Picard-tools (ver. 2.25.0; https://broadinstitute.github.io/picard) to remove duplicate reads; Genome Analysis TK (ver. 4.1.8.0) [17] to rearrange the bam format, extract mutation candidates, identify and filter variants relative to K7, and identify the mutations. Finally, variants were annotated manually with the aid of SnpEff software (ver. 4.3; https://pcingola.github.io/SnpEff/) [18].

### 2.4. Fluorescence Staining, Microscopy, and Image Processing

We cultivated cells of strains K7, K7GE01, K7GE21, K7GE31, and K7GE41 until the early log phase (<5 × 10^6^) and fixed them with medium containing 3.7% (*w/v*) formaldehyde (Wako, Osaka, Japan). We then triple-stained cells with fluorescein isothiocyanate-conjugated concanavalin A (Sigma, St. Louis, MO, USA) for the cell wall, rhodamine-phalloidin (Invitrogen, Carlsbad, CA, USA) for the actin cytoskeleton, and 4′,6-diamidino-2-phenylindole (Sigma) for nuclear DNA, as described previously [19]. We acquired fluorescence microscopy images of the cells using a microscope (Axio Imager) equipped with a special lens (6100 EC Plan-Neofluar; Carl Zeiss, Oberkochen, Germany), a cooled-charge-coupled device camera (CoolSNAP HQ; Roper Scientific Photometrics, Tucson, AZ, USA), and appropriate software (AxioVision; Carl Zeiss). We analyzed the micrographs of the cells with image processing software designed for diploid cells (CalMorph, ver. 1.3) [20]. We obtained the morphological data of 501 traits from the single-cell data. Descriptions of each trait have been presented previously [19]. The CalMorph user manual is available at the *S. cerevisiae* Morphological Database (https://www.yeast.ib.k.u-tokyo.ac.jp/CalMorph/).

### 2.5. Principal Component Analysis (Pca) for Dimensional Reduction and Calculation of the Euclidean Distance in the Degenerated Morphological Space

Morphological data obtained in this study (strains K7, K7GE01, K7GE21, K7GE31, and K7GE41) were used for statistical analyses. There were eight samples for strain K7, and four each for strains K7GE01, K7GE21, K7GE31, and K7GE41, respectively. We applied PCA to the mean Z-values in each strain for all 501 traits that were calculated using a general linear model. From the PCA of the five sake yeasts, the cumulative contribution ratios of the first 11 principal components reached 90%.

The Euclidean distance [21] was used to assess morphological differences between two strains; this distance is near zero if the cell morphology of the two strains is similar, but is otherwise larger. The Euclidean distance between each strain was calculated from the principal component scores of the first 11 principal components (cumulative contribution ratio 90%), as described previously [22]. To calculate the principal component scores of each strain, the Z-values from each independent experiment in each strain were projected onto the 11 principal components. The mean values and standard deviations were calculated from the Euclidean distances of each replicate from the center of the parental strain in the orthogonal morphological space of 11 PCs.

### 2.6. Small-Scale Fermentation Test

A sake mash was prepared by mixing 72.8 g of pregelatinized rice (corresponding to 100 g of white rice), 19.2 g of dried koji (rice with *Aspergillus oryzae* mold, corresponding to 20 g of white rice), 136 μL of 90% lactic acid, and 170 mL of water containing 1 × 10^9^ precultured yeast cells. The mash (three replicates) was incubated at 15 °C for 20 days without shaking. The fermentation was monitored daily by quantifying the amount of evolved CO_2_, measuring the weight loss of the sake mash. After the sake fermentation, the mash was collected in 50-mL centrifuge tubes and centrifuged at 15 °C, 5000 rpm for 15 min. The supernatant was filtered with microfiber cloth to yield the sake product and stored at −80 °C.

### 2.7. Component Analysis of Sake Made in the Fermentation Test

The sake meter value (SMV) was calculated after measuring the density of the sake relative to water. Density was measured with a density/specific gravity meter (DA-650; Kyoto Denshi, Kyoto, Japan) equipped with an autosampler (CHD-502; Kyoto Electronics Manufacturing, Kyoto, Japan). A thawed sample of 10 mL or more was added to a 20-mL vial for measurement. SMV was calculated as SMV = (1/specific gravity − 1) × 1443. A more negative SMV indicates a higher specific gravity and sugar content.

The ethanol concentration was measured with a gas chromatograph (6890 N; Agilent Technologies, Santa Clara, CA, USA) equipped with an auto injector (7683B; Agilent). As an internal standard, 1% isopropanol (960 μL) was mixed with the sample (40 μL).

Aroma components, such as ethyl acetate, 1-propanol, isobutanol, isoamyl alcohol, isoamyl acetate, and ethyl caproate, were measured by a gas chromatograph (6890 N; Agilent) equipped with a head space sampler (HSS 7697A; Agilent). Internal standards (100 μL) were mixed with the samples (900 μL).

Acidity and amino acid contents were measured with an automatic titrator (COM-1700; Hiranuma, Ibaraki, Japan). Measurements were performed with 10 mL of the samples.

Organic acids, including malic acid, succinic acid, lactic acid, citric acid, acetic acid, and phosphoric acid, were measured by a liquid chromatograph (CBM40 equipped with an SCR-102H column; Shimadzu, Kyoto, Japan). A thawed 1 mL sample was injected for measurement.

The precursor of dimethyl trisulfide (1,2-dihydroxy-5-(methylsulfinyl) -pentan-3-one; DMTS-P1) was analyzed by LC-MS (LCMS8040, Shimadzu) using (ethyl-d3)-DMTS-P1 as an internal standard [23,24].

### 2.8. PCA of Sake Components

We performed PCA using 16 or 19 parameters on the mean values yielded by component analysis of 12 samples (four strains × three samples) of sake produced in the small-scale fermentation test. The mean value of each strain, and the value of each sample, were plotted with two components. Among the correlation coefficients obtained by mapping the values of the 12 samples, we considered those with *p* < 0.05 after Bonferroni correction to be statistically significant.

### 2.9. Structural Changes Predicted Based on the N532K Change in Nrd1

The secondary and tertiary protein structures of Nrd1 of the S288C reference genome (Saccharomyces Genome Database; SGD) were predicted by Iterative Threading Assembly Refinement (I-TASSER) [25]. The first ranked PDB files, defined by the C- and TM-scores [26], were visualized by UCFS Chimera [27].

## 3. Results

### 3.1. Isolation of Genome-Edited Sake Yeast Strain with Eight Mutations

The non-foam-forming yeast strain K7GE01, which was constructed with genome editing technology in our previous study, was used as a starting material (Figure 1A). We performed an additional three-step serial breeding procedure with genome editing technology using the improved CRISPR-Cas9 system [12]. The advantages of this system include single-step transformation, the use of dominant selectable markers (G418^r^) suitable for markerless sake yeast isolates, the ability to generate markerless alleles, and the ease of eliminating the genome editing plasmid. By using any of the genome-edited sake yeasts created in this study, the effective capacity of a brewing tank can be increased because *awa1*∆/*awa1*∆ possesses a non-foam-forming characteristic. The second introduced change, *car1*∆/*car1*∆, causes a defect in the urea cycle. Because of this, the genome-edited yeasts after K7GE21 were intended to produce sake with improved safety, containing less ethyl carbamate (Figure 1B). DMTS is hardly generated by *mde1*∆/*mde1*∆, the third introduced change. This change was intended to produce sake that does not yield an unpleasant scent, *hineka*, during storage for strains after K7GE31 (Figure 1C). Finally, because some people like sake with a strong *ginjo* aroma, we introduced point mutations in *FAS2* to generate the *FAS2* (G1250S)/*FAS2* (G1250S) strain (Figure 1D). The final genome-edited strain, K7GE41, comprises the standard sake yeast strain, K7, with the eight mutations *awa1*∆/*awa1*∆, *car1*∆/*car1*∆, *mde1*∆/*mde1*∆, and *FAS2* (G1250S)/*FAS2* (G1250S). Comparison of whole-genome sequences revealed that one heterozygous mutation and one loss of heterozygosity (LOH) were introduced into the genome during four steps of the breeding process (Appendix A).

K7GE21, K7GE31, and K7GE41 each contained a single heterozygous mutation corresponding to N532K of Nrd1, an essential gene encoding the RNA-binding subunit of the Nrd1 complex (Appendix A). As this mutation was not present in the parental K7GE01 strain, it probably appeared during transformation, cultivation, or storage of K7GE21. This mutation was heterozygous and Nrd1 N532K caused no obvious changes in its predicted protein structure in this region. This mutation is far downstream of the functional domains of Ndr1, such as the CTD-interacting domain (62–137 aa) and RNA recognition motif (341–396 aa). These results suggested that the effect of the heterozygous off-target mutation in K7GE21 was negligible.

K7GE31 and K7GE41 harbored the same homozygous SNPs due to LOH. The corresponding heterozygous SNPs pre-existed in the original K7 isolates (K7GE01 and K7GE21), implying that a single LOH was introduced when K7GE31 was made. Although it is not known how LOH occurs, it has been observed at high frequency in sake yeast isolates [2]. The genome-edited *MDE1* and LOH locus were very close together (~70 kb) on chromosome X, suggesting that the CRISPR/Cas9-dependent double strand break induces LOH.

### 3.2. Genome Editing Efficiency of Sake Yeast Strains

PCR was used to investigate whether the mutations had been introduced during breeding as intended. Genome editing efficiency was calculated as the number of yeast transformants into which the intended mutation had been introduced, divided by the number of transformants examined. This efficiency was highest in the *CAR1* locus, in which 96 ± 4% of transformants were genetically modified correctly (*car1*∆/*car1*∆; Figure 2). Genome editing efficiency was lowest in the *AWA1* locus, in which only 16 ± 3% of transformants were genetically modified correctly (*awa1*∆/*awa1*∆; Figure 2). Homozygous mutations were introduced in all strains, with no heterozygous mutations. These results suggested that homozygous mutations are always introduced during the genome editing of sake yeast, and that genome-editing efficiencies were dependent on the genetic locus.

### 3.3. Absence of Urea Production in Sake Brewed with car1∆/car1∆ Yeast Strains

The potential carcinogen ethyl carbamate is converted from urea during the storage of sake [7]. Urea is produced from arginine by arginase in the urea cycle. Therefore, we deleted the *CAR1* gene encoding arginase to breed yeast that makes sake with fewer safety issues. We created the following recombinant sake yeasts with homozygous *car1*∆/*car1*∆ mutations: K7GE21, K7GE31, and K7GE41 (Figure 1).

To confirm the properties of sake made with the genome-edited *car1*∆/*car1*∆ strains, we analyzed the components of sake made in small-scale fermentation tests. As expected, the sake brewed with these strains contained significantly lower levels of urea than the sake made with their parent *CAR1*/*CAR1* strain, K7GE01 (Tukey’s multiple comparison test, *p* < 0.05; Figure 3A). Their low levels of urea were almost equivalent to those made with the existing non-urea-producing sake strains, Karg7 and K1901 [28], suggesting that our genome-edited *car1*∆/*car1*∆ strains can be used in practice as non-urea-producing sake yeast.

### 3.4. Reduction of a DMTS Precursor in Sake Brewed with mde1∆/mde1∆ Yeast Strains

After extended storage of sake, especially at a relatively high temperature, DMTS, a major unpleasant component of *hineka*, is derived from its precursor DMTS-P1. DMTS-P1 is produced during sake fermentation by the enzymatic action of Mde1 and Mri1, which are involved in the methionine salvage pathway [29].

Therefore, sake yeast defective in the *MDE1* gene exhibits decreased production of DMTS [29]. In this study, we deleted the *MDE1* gene to breed sake yeast that makes sake with a less unpleasant aroma. We created the following recombinant sake yeasts with homozygous *mde1∆/mde1∆* mutations: K7GE31 and K7GE41 (Figure 1).

By small-scale fermentation tests, we measured the concentrations of the precursor DMTS-P1 in sake made with several genome-edited strains. As expected, the sake made with the *mde1∆/mde1∆* strains contained significantly lower levels of DMTS-P1 than the sake made with their direct ancestor *MDE1/MDE1* strains, K7GE01 and K7GE21 (Tukey’s multiple comparison test, *p* < 0.05; Figure 3B). The levels of DMTS-P1 decreased 20-fold, suggesting that our genome-edited *mde1∆/mde1∆* strains can be used to brew sake with a less unpleasant aroma.

### 3.5. Enhanced Ethyl Caproate Content in Sake Brewed with FAS2 (G1250S)/FAS2 (G1250S) Yeast Strain

The *ginjo* aroma is caused by ethyl caproate, which is produced from ethanol and caproic acid. Caproic acid is synthesized from acetyl-CoA, malonyl-CoA, and nicotinamide adenine dinucleotide phosphate in the fatty acid synthesis pathway. Therefore, the Gly1250Ser mutation in the *FAS2* gene, which encodes the alpha subunit of fatty acid synthase, enhances the production of ethyl caproate. Accordingly, we introduced this mutation to breed sake yeast that produces sake with a strong *ginjo* aroma. We created the following recombinant sake yeast strain with homozygous *FAS2* (G1250S)/*FAS2* (G1250S) mutations: K7GE41 (Figure 1).

With small-scale fermentation tests, we measured the concentrations of ethyl caproate in sake made with several genome-edited strains. We found that the sake made with the *FAS2* (G1250S)/*FAS2* (G1250S) strain contained significantly higher levels of ethyl caproate than the sake made with its direct ancestor strains, K7GE01, K7GE21, and K7GE31 (Tukey’s multiple comparison test, *p* < 0.05; Figure 3C). The concentration of ethyl caproate produced by K7GE41 was 4.5-fold greater than that produced by its parent strains and was equivalent to that produced by the standard ethyl caproate-producing strain, K1801 [30]. These results suggested that the final genome-edited yeast strain K7GE41 produces sake with a strong *ginjo* aroma.

### 3.6. Fermentation Activities of the Genome-Edited Strains

We conducted small-scale fermentation tests to investigate the fermentation characteristics of the genome-edited strains in sake brewing. As an indicator of fermentation activity, we measured the CO_2_ emission levels deduced from the weight of the fermentation mash. According to the reaction equation, theoretically, CO_2_ emission and ethanol production occur equally. The daily CO_2_ emission levels peaked at 4 days for all strains examined (Figure 4A). However, we observed a slow start-up of fermentation with K7GE41, the strain that produces sake with a strong *ginjo* aroma. Comparing the cumulative CO_2_ emissions up to the peak (day 4), the amount was significantly lower for K7GE41 than the others (Tukey’s multiple comparison test, *p* < 0.05; Figure 4B). This led to a delay in the reduction of postpeak fermentation with this strain (Figure 4A), suggesting that fermentation with K7GE41 was slightly delayed in the early stages of sake brewing. We did not detect any differences in total CO_2_ emission at the end of fermentation (Tukey’s multiple comparison test at the 0.05 significance level; Figure 4B), indicating that these genome-edited strains had similar ethanol production (Appendix A).

### 3.7. Component Analysis of Sake Made in the Fermentation Test

To reveal the characteristics of sake made by the genome-edited strains, we analyzed the standard 19 components of sake. Because the fermentation kinetics showed differences among strains (Figure 4A), as described in the previous section, we expected to observe differences in sake components between *K7GE41* and other strains. First, to visually characterize the differences, we examined all strains in the degenerated orthogonal space after performing PCA with the components of sake made in small-scale fermentation tests. We performed two types of PCA, namely with and without the three changed, targeted components (urea, DMTS-P1, and ethyl caproate). In both instances, K7GE41 was located substantially far from the other strains in the first principle component–second principle component (PC1–PC2) orthogonal space (Figure 5A,B). When we performed PCA with the 16 components (after removal of the three purposely changed components), the contributions of PC1 and PC2 accounted for 69% and 15% of the variance, respectively (Figure 5B), and the cumulative contribution was 84% (Appendix A). The genome-edited strains, K7GE01 (yellow dot), K7GE21 (green dot), and K7GE31 (red dot) lie together at the left-hand side in this degenerated space. In contrast, strain K7GE41 (blue dot) lays at the right-middle, almost parallel to the direction of fermentation-related components such as pyruvate and SMV (red arrows), aroma components such as ethyl acetate and isoamyl acetate (grey arrows), and amino acidity (blue arrow).

The results deduced from the PCA mentioned above were confirmed by direct statistical comparison of the individual components. Considering the fermentation-related components, sake made from K7GE41 had significantly higher pyruvate (Tukey’s multiple comparison test, *p* < 0.01; Figure 6A) and significantly lower SMV (*p* < 0.05; Figure 6B) than did sake made from K7GE01. Aroma components such as ethyl acetate and isoamyl acetate were significantly decreased in sake made from K7GE41 (*p* < 0.05; Figure 6C,D). Amino acidity significantly increased in sake made from K7GE41 (*p* < 0.05; Figure 6E). In Appendix A, we summarized other components that unexpectedly changed for K7GE41. Compared with the 14 components that changed during the breeding of K7GE41, fewer changes were observed in other breeding processes. When *car1*∆/*car1*∆ was introduced into K7GE01 to make K7GE21, only four components (isoamyl alcohol, 1-propanol, succinic acid, and lactic acid) were changed significantly and unintentionally (*p* < 0.05; Appendix A). No unexpected components were changed during the breeding of K7GE31. Ethanol, acidity, and malic acid did not change during the serial breeding steps (Appendix A). Thus, many components changed unintentionally during the serial breeding of these sake yeast strains.

### 3.8. Morphological Analysis of Genome-Edited Sake Yeast Strains

We previously reported the morphological diversity of sake yeast isolates in the K7-lineage [22]. Remarkable morphological changes were also observed when non-foam-forming yeast strains were generated with genome editing technology. Therefore, this study investigated what kind of morphological changes occurred during the serial breeding steps. High-dimensional morphological phenotyping of K7, K7GE01, K7GE21, K7GE31, and K7GE41 was performed using the CalMorph image analysis system after staining the cell wall, actin, and nuclear DNA. In the two-dimensional morphological space, this clarified that each of the four breeding stages showed characteristic morphological changes (Figure 7A). Large morphological changes were observed during the step generating K7GE41.

The holistic morphological abnormality (HMA) is a measure of how large a morphological change is [31]. Examination of HMA at each step revealed that the largest morphological change occurred when K7GE41 was generated (*p* < 0.05 after the Bonferroni correction, likelihood ratio test of one-way ANOVA with a gamma distribution, Figure 7B). When K7GE31 was generated, there was no detectable change in HMA (Figure 7B). The morphological changes at each step were detected with the likelihood ratio test (FDR = 0.01, Appendix A). The clearest morphological changes were detected during the breeding of K7GE41. The K7GE41 cells became obviously larger (Figure 7C), in good agreement with the observation that the sake yeast strain producing a sweet *ginjo* aroma was large [32]. In addition, K7GE41 exhibited morphological changes, such as an increased proportion of cells with delocalized actin patches, increased G2 cells, and nuclear localization in mother cells (Figure 7C). This was thought to be the result of the FAS2 (G1250S) mutation. We concluded that various characteristic morphological changes occurred due to the mutations introduced during the four steps of the breeding process.

## 4. Discussion

We applied genome editing technology to sake yeast isolates in a previous study [8] with the aim of performing parallel breeding by deletion of *AWA1* in strains K6, K7, K9, and K10. In the present study, we attempted to introduce *car1*∆/*car1*∆, *mde1*∆/*mde1*∆, and *FAS2* (G1250S)/*FAS2* (G1250S) mutations into a K7-derived diploid non-foam-forming sake yeast strain (*awa1*∆/*awa1*∆) by means of genome editing technology. We finally created a serial breeding strain (K7GE41) that contained eight mutations. Analyses of the components of the sake made with the genome-edited strains revealed that urea, the precursor of ethyl carbamate, was dramatically reduced as expected for *car1*∆/*car1*∆; DMTS-P1, the precursor of *hineka* (DMTS), was hardly detected as expected for *mde1*∆/*mde1*∆; and a large amount of ethyl caproate was produced as expected for *FAS2* (G1250S)/*FAS2* (G1250S). Although serial breeding is often considered to be laborious and time-consuming, this study is the first successful attempt to generate a sake yeast strain with these four excellent brewing characteristics. Our findings demonstrate that genome editing technology is extremely effective for the breeding of sake yeast strains.

### 4.1. Variation of Genome Editing Efficiency during Sake Yeast Breeding

After sake yeast breeding, we found that the genome editing efficiencies at the *AWA1*, *CAR1*, *MDE1*, and *FAS2* loci varied from 16% to 96%. Previous studies have shown that the efficiency and specificity of genome editing are affected by the host strain, CRISPR-Cas9/sgRNA system to express ribozyme, and sgRNA design [8]. In particular, unlike laboratory strains such as S288C, the choice of tRNA promoter used for the expression of the ribozyme was important in the polyploid industrial S. cerevisiae strain ATCC4124 for improving tolerance and productivity [12]. We performed all genome editing of sake yeast using the same pCas9-Pro-based plasmid harboring the tRNAPro promoter used in ATCC4124 [12]. Because a maximum genome editing efficiency of 96% was achieved, we consider there to be sufficient compatibility between the tRNAPro and sake yeast. For the design of sgRNA, we consistently followed the method described by Ohnuki, et al. [8] but the use of an alternative protospacer adjacent motif sequence may change the genome editing efficiency. Furthermore, genes present in the heterochromatin and nucleosome regions tend to be less efficient in genome editing [33,34]. The *AWA1* gene, which exhibited the lowest genome editing efficiency, is in the heterochromatin region near the telomere region on chromosome XV. Furthermore, regarding *AWA1*, dynamic genomic changes were observed in almost all transformants, although many colonies did not have the correct gene deletion, which implied that the cleavages were rarely repaired in the desired manner. This may be due to the fact that the *AWA1* gene deletion was made after the deletion of a large (6 kb) genomic fragment. Another notable feature during the genome editing of sake yeast strains was the appearance of a high frequency of homozygous mutations, such that we observed no heterogeneous mutations in more than 400 transformants. This could be caused by strong tRNAPro promoter activity for ribozyme expression, or by the loss of homozygosity frequently observed in the sake yeast genome [2].

### 4.2. Changes in the Components of Sake due to FAS2 (G1250S) and car1∆

Sake made by yeast isolates with the *FAS2* (G1250S)/*FAS2* (G1250S) mutation was previously shown to contain a large amount of ethyl caproate. However, it was unclear whether those yeast isolates harbor any extra mutations. By comparing the genome-edited strains K7GE41 and K7GE31, we have elucidated the characteristics of sake affected by *FAS2* (G1250S)/*FAS2* (G1250S). The sake made with K7GE41 had a high concentration of pyruvic acid and a low SMV (indicating glucose retention). In our small fermentation test, K7GE41 exhibited a slow fermentation rate, which may lead to delayed glucose consumption and the accumulation of intermediate metabolites such as pyruvic acid. Sake made with K7GE41 contained elevated and reduced amounts of ethyl caproate and ethyl acetate, respectively. Because these two aromatic compounds are synthesized by the same ethyl esterase, encoded by *EEB1*, competitive inhibition may have contributed to these changes. Another aroma component, isoamyl acetate, was also reduced in sake made by K7GE41. A previous study [35] also showed an inverse relationship between isoamyl acetate and ethyl caproate, so negative feedback regulation might occur between the isoamyl acetate and ethyl caproate biosyntheses. Finally, the sake made by K7GE41 contained more amino acids. *FAS2* encodes a catalytic subunit of fatty acid synthase, so the introduction of the *FAS2* (G1250S) mutation presumably affects the membrane, thereby increasing the proportion of autolyzed yeast in mash. Indeed, the proportion of viable K7GE41 in the mash was lower than the corresponding proportions of other strains (data not shown). Therefore, autolysis of K7GE41 may cause intracellular amino acid release into the mash.

Deletion of the *CAR1* gene encoding arginase in strains K7GE21, K7GE31, and K7GE41 unexpectedly caused a reduction of isoamyl alcohol, an aroma component synthesized from leucine via keto acid. Although the detailed mechanism of metabolic regulation is unknown, the amount of leucine-derived isoamyl alcohol may have decreased due to the effect of reduced arginine metabolism. LOH was introduced into K7GE31, but there were no obvious holistic morphological changes or component changes of sake during this step. This implies that the phenotypic changes due to this LOH were negligible. Overall, our study revealed unexpected links between the modified genes and brewing properties. By applying genome editing technology to sake yeast strains, our findings have clarified several characteristics of sake affected by *FAS2* (G1250S) and *car1*∆.

### 4.3. Effectiveness of Genome Editing in Serial Breeding

We generated strains K7GE21, K7GE31, and K7GE41 by serial breeding in this study, based on the previously constructed strain K7GE01. A novel genome editing technology enables simultaneous manipulation of multiple genes [12], but this technique requires storage of intermediate sake yeast strains produced by serial breeding. The intermediate strains can be utilized for brewing to make a variety of sake. We first constructed strain K7GE01 because societal demands require improved productivity. We then created strain K7GE21 because safety is widely regarded as important. Some people like the *ginjo* aroma but others do not, so we made strain K7GE31 without *hineka*, and finally constructed strain K7GE41. These varieties generated by genome editing can be chosen according to the desired properties of sake. It is possible to breed sake yeast conventionally with similar, excellent brewing characteristics, but breeding in this manner is laborious and time-consuming. In addition, many off-target mutations may accumulate, and there is a risk that the resulting yeast will exhibit unexpected brewing characteristics.

### 4.4. Breeding of Sake Yeast in the Future

We employed evidence-based breeding using genome editing technology in this study, using our accumulated knowledge of sake brewing. K7GE41 is an optimal sake yeast strain with ideal brewing characteristics, and the sake yeast strains generated during this breeding process are also useful for making a wide variety of sake. This technology will enable the creation of a wider variety of sake yeast by changing the yeast strains initially used. Strains K6, K9, and K10 are from the same lineage of sake yeasts as strain K7, but their brewing characteristics are distinct. Therefore, it may be useful to generate genome-edited yeast strains from each sake yeast. In addition to the yeast strains distributed by the Brewing Society of Japan, the same genome editing technology could be applied to the strains owned by each prefecture and sake brewery. Furthermore, genome editing technology can be applied to design or anticipate changes in metabolic pathways during sake yeast. A long-term goal of sake yeast breeding is to create a strain that exhibits enhanced ethanol resistance. It would also be useful to create yeast that produces high levels of both ethyl caproate and isoamyl acetate. The regulations regarding genome-edited food and beverages have been amended in many countries including Japan. If a foreign gene is not inserted, it is now possible to sell genome-edited food and beverages without a safety review in Japan by simply making a voluntary notification. Therefore, the breeding of sake yeast using genome editing may be actively pursued in the future, and both evidence- and prediction-based breeding of sake yeast are expected to be actively explored.

## 5. Conclusions

We introduced eight mutations into standard sake yeast to construct a non-foam-forming strain that makes sake without producing carcinogens or an unpleasant odor, instead producing a sweet *ginjo* aroma. The few unexpected genetic perturbations introduced during breeding proved that genome editing technology is extremely effective for serial breeding of sake yeast.

## Figures and Tables

**Figure 1 cells-10-01299-f001:**
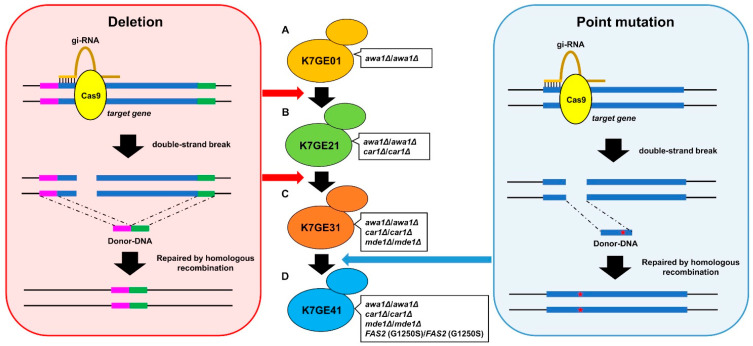
Serial breeding of a sake yeast strain with eight mutations. A double-strand break was introduced in the target gene by CRISPR-Cas9. Homologous recombination repair was performed using donor DNA (150 bp) to introduce gene deletion (light red region) or point mutation (light blue region). Starting from K7GE01 (yellow) with *AWA1* deletion (**A**), K7GE21 (green) with *CAR1* deletion (**B**), K7GE31 (orange) with *MDE1* deletion (**C**), and K7GE41 (blue) with *FAS2* (G1250S) point mutation (**D**) were generated successively. The final K7GE41 sake yeast strain harbored homozygous eight mutations that confer excellent brewing characteristics.

**Figure 2 cells-10-01299-f002:**
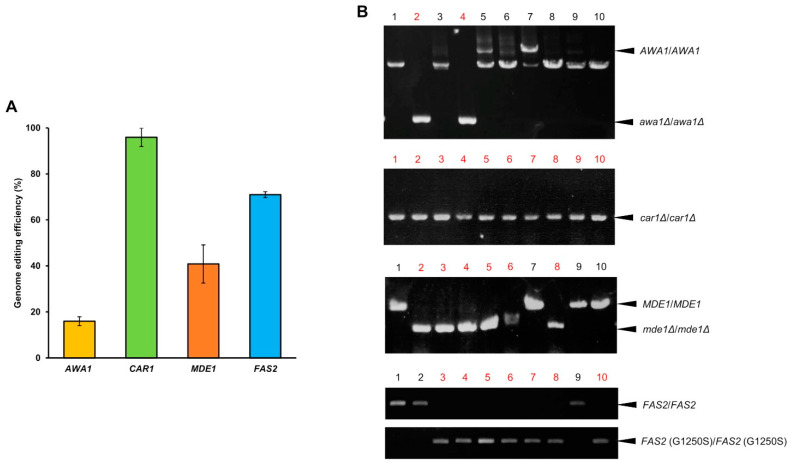
Genome editing efficiency of sake yeast strains. (**A**) Genome editing efficiency at the *AWA1*, *CAR1*, *MDE1*, and *FAS2* loci. Error bars indicate standard error (*n* = 3). (**B**) DNA fragments (*n* = 10 samples each) amplified by colony PCR are shown as examples. The numbers of yeast transformants with correct deletion (*AWA1*, *CAR1* and *MDE1*) or mutation *FAS2* (G1250S) are shown in red.

**Figure 3 cells-10-01299-f003:**
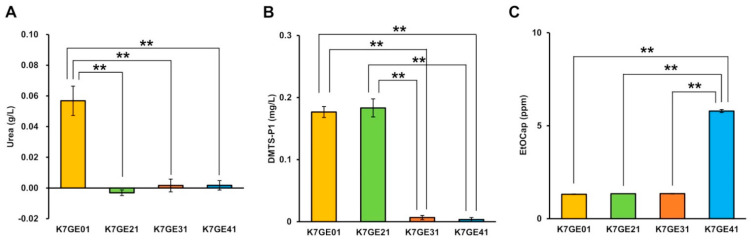
Measurement of urea (**A**), dimethyl trisulfide-P1 (DMTS−P1) (**B**) and ethyl caproate (**C**) in sake made with genome-edited strains. Error bars indicate standard error (*n* = 3). Asterisks indicate Tukey’s multiple comparison test (** *p* < 0.01).

**Figure 4 cells-10-01299-f004:**
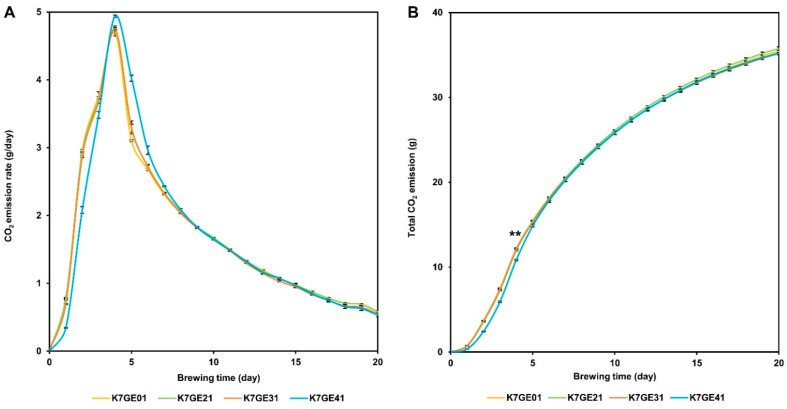
Carbon dioxide (CO_2_) emissions of genome-edited sake strains. Daily (**A**) and total (**B**) CO_2_ emissions measured by weight. K7GE01 (yellow), K7GE21 (green), K7GE31 (orange), and K7GE41 (blue) were examined. Error bars indicate standard error (*n* = 3).** indicates a peak of CO_2_ emission rate (day 4).

**Figure 5 cells-10-01299-f005:**
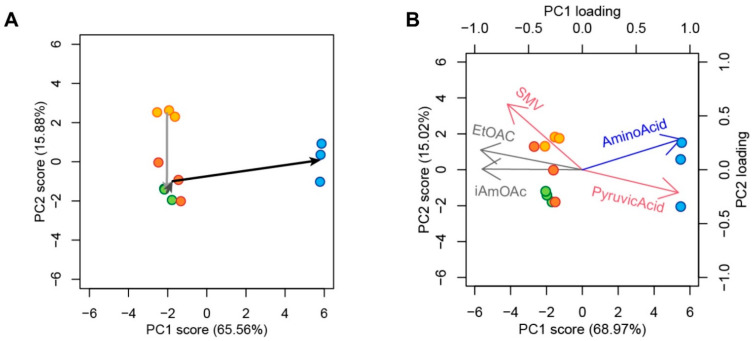
Principal component analysis (PCA) of sake produced by genome-edited yeast strains. Distribution of genome-edited yeast strains in the PC1–PC2 orthogonal space after performing PCA with 19 components (**A**). K7GE01 (yellow), K7GE21 (green), K7GE31 (orange), and K7GE41 (blue) were generated in that order. Distribution of sake components in the PC1–PC2 space after performing PCA with 16 components (without the three intentionally changed components) (**B**). Pyruvate and sake meter value (SMV) (red), ethyl acetate and isoamyl acetate (grey), and amino acidity (blue) are shown as arrows.

**Figure 6 cells-10-01299-f006:**
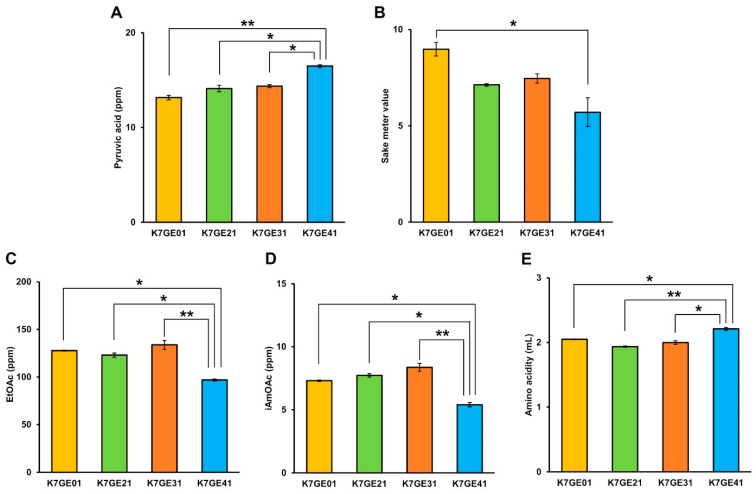
Measurement of pyruvate (**A**), sake meter value (**B**), ethyl acetate (**C**), isoamyl acetate (**D**), and amino acidity (**E**) in sake made with genome-edited strains. Error bars indicate standard error (*n* = 3). Asterisks indicate Tukey’s multiple comparison test (* *p* < 0.05, ** *p* < 0.01).

**Figure 7 cells-10-01299-f007:**
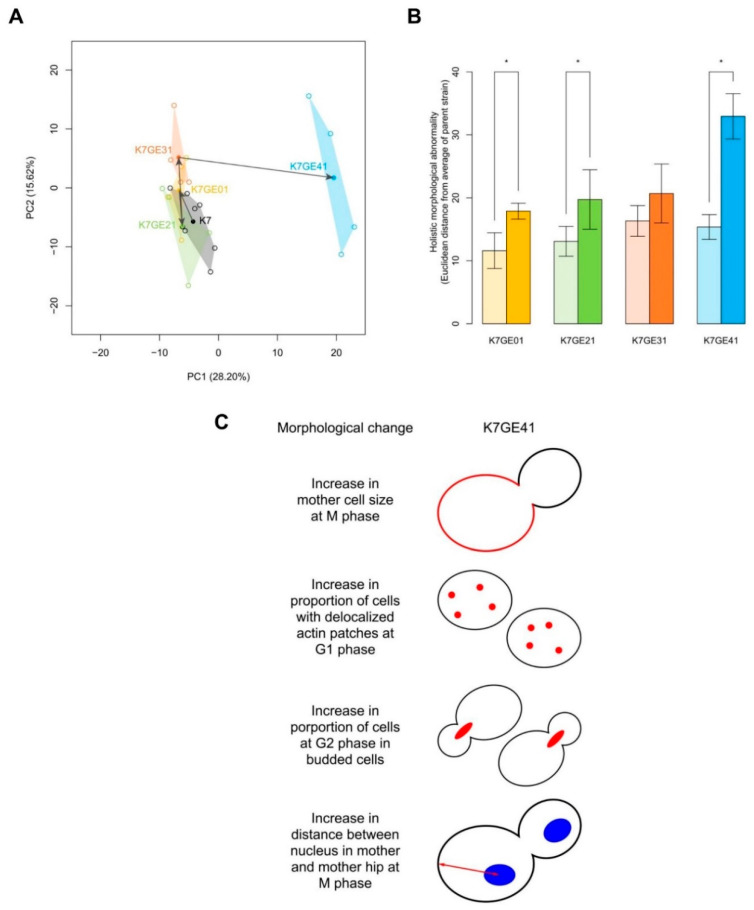
Morphological changes during serial breeding. (**A**) Distribution of sake yeast isolates in the orthogonal morphological space. Principal component analysis was performed on 501 morphological traits of four genome editing strains. K7GE01, K7GE21, K7GE31, and K7GE41 (*n* = 4 each, open circles) and the standard K7 strain (*n* = 7, open black circles). The colors of the strain names and regions are the same as in Figure 1. The first two PCs (PC1 *x*-axis, PC2 *y*-axis) captured 43.82% of the variance, as shown in parentheses. The center of each strain was calculated as the average of the PC score and is shown as a solid circle. Arrows indicate the direction of breeding. (**B**) Euclidean distances from their parental strain. The holistic morphological abnormality was calculated as the Euclidean distance of each replicate from the center of the parental strain in the orthogonal morphological space of the 11 PCs capturing 90% of the variance of the 501 traits. Light and vivid colors indicate the parent strains and genome-editing strains K7GE01, K7GE21, K7GE31, and K7GE41, respectively. The error bar indicates the standard deviation. An asterisk indicates a significant difference at *p* < 0.05 after the Bonferroni correction (likelihood ratio test of one-way ANOVA with a gamma distribution). (**C**) Schematic representation of the morphological features of the genome-edited sake yeast strains. One-way ANOVA of the generalized linear model was applied to 501 morphological traits of five sake yeast strains, and 31 traits were detected at *p* < 0.05 after the Bonferroni correction. Of the 31 traits, 23 in at least one genome editing strain differed significantly from K7 at *p* < 0.05 by the Wald test. K7GE01, K7GE21, K7GE31, and K7GE41 differed significantly from K7 in 4, 3, 9, and 17 traits, respectively. The 23 detected morphological traits were summarized by successive PCA with 114 replicates of BY4743 and are illustrated.

## Data Availability

Any additional data will be available upon request to the corresponding author.

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
