# Peer review of "Genome Editing to Generate Sake Yeast Strains with Eight Mutations That Confer Excellent Brewing Characteristics"

_cells, 2021, doi:10.3390/cells10061299_

Round 1
Reviewer 1 Report
Line 36: change sake yeast for the yeast
Line 78: japanese is not necessary
Line 164: % of formaldehyde
Line 293 Homozygous mutations were introduced in all strains, with no heterozygous mutations. Any idea of why?
Figure 4A: it would be helpful to calculate the AUC of the CO2 and perform statistics
Author Response
We appreciated the reviewer#1’s valuable comments.
As for the absence of heterozygous mutations during genome editing breeding, it is an interesting observation, and we are now studying its mechanism. This could be related to frequently observed LOH (loss of heterozygosity) events in sake yeast. Combined effect of ribozyme expressed in sake yeast may also be the case. We discuss these possibilities in the Discussion (Lines 502-506).
Another notable feature during the genome editing of sake yeast strains was the ap-pearance of a high frequency of homozygous mutations, such that we observed no het-erogeneous mutations in more than 400 transformants. This could be caused by strong tRNAPro promoter activity for ribozyme expression, or by the loss of homozygosity frequently observed in the sake yeast genome [2]. (502-506)
As for your comments on Figure 4A, Area Under the Curve (AUC) is shown in Figure 4B. We performed statistical analysis (Tukey's multiple comparison test) on AUC before the peak and added the sentence (Lines 370-372).
Comparing the cumulative CO2 emissions up to the peak (day 4), the amount was significantly lower for K7GE41 than the others (Tukey's multiple comparison test, p < 0.05; Figure 4B). (370-372)
According to the reviewer‘s suggestions, we corrected typo errors and added missing information (Lines 36, 78, 168).
Reviewer 2 Report
I would like to draw the authors' attention to the following few things:
In section 2.6, please indicate the number of replicates of small-scale fermentations with each yeast strain.
In section 3.7, line 375, the authors write strain ‘K7GE04’. This is certainly a typo and the 'K7GE41' strain was thought of.
In FigS1 (A), the notation 'vol %' is more accurate for alcohol content.
In the case of FigS1 (B), the unit of measurement for 'Acidity' is 'mL', this may be a misconception, may be the authors meant g/L?
Author Response
We appreciated the reviewer#2’s valuable comments.
According to the suggestion, we added the information of replicates in section 2.6. (Line 206)
We changed from K7GE04 to K7GE41 in Line 385.
We changed to the notation “vol %” in Fig. S1(A).
We added the explanation of “acidity value” in the legend of Fig. S1(B) as shown below.
Acidity value indicates volume of 0.1N NaOH required for neutralization of sake samples.
Reviewer 3 Report
The manuscript is a high-quality work both in the presentation of the documentation and in the scientific content. I do not have any suggestion to improve the manuscript because it is very-well written.
The theme is very interesting, with novelty, inserting advanced technology to obtain strains for sake production.
Author Response
We appreciated the reviewer#3’s high evaluation on this study.
Reviewer 4 Report
This manuscript succeeded to constuct a a non-foam-forming strain with eight mutations that makes sake without producing carcinogens or an unpleasant odor.Publication of this work in cells can be considered after the authors address the following concerns.
1 The world "octuple mutations" is not suitable. The final strain (K7GE41) contained eight mutations rather octaploid. So, "multiple mutations" maybe better.
2 During the small-scale fermentation tests, ethanol concentration should also be measured.
3 Why the Gly1250Ser mutation in the FAS2 gene make the sake yeast strains producing the strong ginjo aroma of ethyl caproic acid? Is any difference of the fatty acid synthesis pathway in the mutations?
Author Response
We appreciated the reviewer#4’s valuable comments.
1.According to the suggestion, we changed all “octuple” to “eight”.2. We measured the final ethanol concentration as shown in Figure S1(A).
As for the fermentation activity during the small-scale fermentation test, we monitored CO2 emission instead of ethanol concentration as described in Figure 4A and B. According to the reaction equation, theoretically, CO2 emission and ethanol production is the same, meaning that ethanol changes like curve in Figure 4B during fermentation. We added this information in the text as shown below (Line 365-368).
We did not directly check ethanol concentration during the fermentation process, because sampling will change the fermentation conditions, even if it is a small amount. In addition, moromi is a complex mixture containing fermented rice, fungi and yeast, etc. So, it is difficult to take a uniform sample during fermentation.
As an indicator of fermentation activity, we measured the CO2 emission levels deduced from the weight of the fermentation mash. According to the reaction equation, theoreti-cally, CO2 emission and ethanol production occur equally. (Line 365-368)
3. As for FAS2 (G1250S), this dominant FAS2 mutation is thought to decrease the activity of the fatty acid synthase complex.
Decreased carbon chain elongation activity by this mutation during fatty acid synthesis increased the amount of caproic acid and Caproyl CoA, precursors of ethyl caproate, and therefore probably results in the increase of ethyl caproate. We added this information in Introduction (Line 87-90).
This dominant FAS2 mutation was thought to decrease fatty acid synthesis, resulting in decreased carbon chain elongation and increased amounts of caproic acid and caproyl-CoA, precursors of ethyl caproate [10]. (87-90).
Round 2
Reviewer 4 Report
This manuscript can be accepted.